# Morphogenesis of *Fractofusus andersoni* and the nature of early animal development

Frances. S. Dunn [1] ✉, Philip C. J. Donoghue [2] & Alexander G. Liu [3]

Rangeomorphs are among the oldest anatomically complex macroscopic fossil organisms and, originating prior to 574 Ma, they represent the earliest total-group eumetazoans. Rangeomorph morphogenesis is therefore significant for understanding the early diversification of eumetazoan bodyplans. However, previous analyses of rangeomorph development have focused on uniterminal forms (possessing only one frond), leaving biterminal and multiterminal rangeomorph bodyplans poorly understood. We describe a population of the biterminal rangeomorph *Fractofusus andersoni* from the Mistaken Point Ecological Reserve UNESCO World Heritage Site of Newfoundland, Canada, and construct a model of growth in *F. andersoni* that rationalises variation between *Fractofusus, Charnia, Bradgatia* and other rangeomorphs, providing a framework for explaining evolutionary transitions between the bodyplans of these members of the eumetazoan stem-group. Our results imply that complex developmental regulatory machinery was already being utilised during the late Ediacaran in the earliest-diverging eumetazoan taxa represented in the fossil record.

The late Ediacaran Period (~579–539 million years ago [Ma]) was a time of profound change in the Earth system, characterised by the first appearance of diverse animal communities[1–3]. Deep-marine palaeocommunities from strata of this age are dominated by fossils of frondose taxa, described as either arboreomorphs or rangeomorphs[4,5]. Rangeomorphs appear prior to 574 Ma[6], are distinguished by their self-similar branching architecture[7,8], and are increasingly interpreted as stem-group eumetazoans (e.g.[9,10]). Rangeomorphs are morphologically varied (Fig. 1), but their intra-relationships remain contentious[11,12].

The morphology of *Fractofusus* Gehling and Narbonne 2007[13] is unlike that of any other described rangeomorph genus (Fig. 1), exhibiting a spindle-shaped form composed of two rows of clustered rangeomorph branches emanating from a shared central axis (ref. 8,13 though see ref. 14). *Fractofusus* thus represents an ideal model system in which to establish morphogenetic principles from which the development of other rangeomorphs may be rationalised, providing a framework for discriminating among competing hypotheses for the generation of rangeomorph form and developmental evolution more broadly. This is significant because previous studies have suggested that step-changes in developmental regulation presage the diversification of animal life during the early Cambrian Period (e.g.[15]). The ichthyosporean, filasterean and choanoflagellate relatives of animals possess orthologues of genes implicated in the development of animal traits (e.g. multicellularity[16,17]), but have not evolved beyond the intercalation of a multicellular stage in their otherwise unicellular life cycles[18]. An often-cited reason for this apparent limitation is the simplicity of their underlying developmental regulation, which plays out temporally, rather than temporally and spatially as in animals[19,20] (though see ref. 21). The evolution of developmental regulation is therefore widely implicated in a number of key transitions, including correlations between the origin of animals and distal regulation[16,22] and the diversification of bilaterians and distal enhancer elements[23]. Understanding such step changes requires knowledge of developmental pattern in animal groups which subtend major nodes in the animal tree. For Eumetazoa, such knowledge can be obtained by understanding the morphogenetic basis of distinct rangeomorph morphologies.

[1]Oxford University Museum of Natural History, Parks Road, University of Oxford, Oxford OX1 3PW, UK. [2]Bristol Palaeobiology Group, School of Earth Sciences, Life Sciences Building, Tyndall Avenue, University of Bristol, Bristol BS8 4QQ, UK. [3]Department of Earth Sciences, Downing Street, University of Cambridge, Cambridge CB2 3EQ, UK. ✉e-mail: frances.dunn@oum.ox.ac.uk

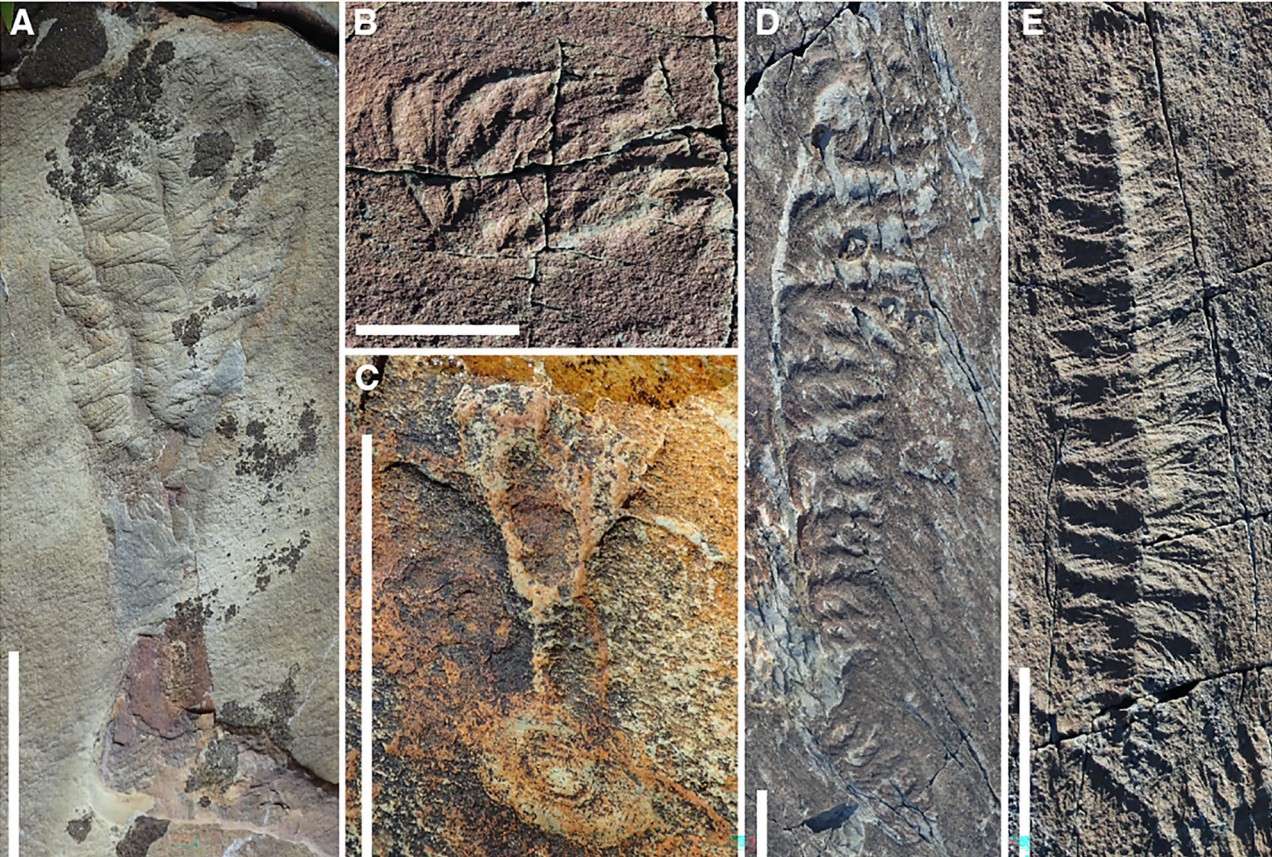

**Fig. 1 | The diversity of rangeomorph bodyplans. A** *Culmofrons plumosa*, field specimen from the MUN surface, Discovery Global Geopark. **B** *Bradgatia* sp. field photograph from the Mistaken Point E surface, Mistaken Point Ecological Reserve (MPER). **C** ?*Primocandelabrum* sp. from the Brasier Surface, MPER. **D** *Pectinifrons abyssalis* from Mistaken Point North, MPER. **E** *Fractofusus misrai* from the Mistaken Point E surface, MPER. Scale bars = 5 cm.

Previous work has suggested that rangeomorphs regulated their bodyplans only minimally (e.g.[24]) implying a later assembly of complex regulatory machinery, perhaps inspired by hypotheses of plasticity in the bodyplans of rangeomorphs[25]. In contrast, highly-regulated bodyplans in rangeomorphs, before the origin of guts and perhaps muscles, would suggest a lag between developmental complexity and morphological innovation, as has previously been suggested for the Bilateria[15]. We describe a population of *Fractofusus andersoni* Gehling and Narbonne 2007[13] from the Mistaken Point Ecological Reserve (MPER) UNESCO World Heritage Site of Newfoundland, allowing us to investigate how morphogenesis explains variation in the morphology of *Fractofusus* and other rangeomorphs. This dataset encompasses the smallest specimens of this taxon yet documented. We use quantitative analysis to construct a model of morphogenesis for *F. andersoni* that is compatible with that of other rangeomorphs, informing an evolutionary model that rationalises variation between multiple rangeomorph taxa. This model provides a framework in which to explain bodyplan transitions across Rangeomorpha, and thus inform debate concerning the evolution of developmental regulation during the diversification of animal bodyplans.

## Results

*Fractofusus andersoni* is characterised by first order branches deriving in turn, from one vane followed by the other, from a central axis. These branches decrease in length towards both distal tips and are each constructed of a single frondlet (as opposed to a bundle of frondlets, as is the case in *F. misrai*). The midline is a narrow, linear structure, typically expressed in positive epirelief. At least 15 specimens from our population are incomplete, terminating abruptly along a line

perpendicular to the long axis of the specimen (Fig. 2A–F; Supplementary Fig. 1, Supplementary Dataset 1, also noted in *F. mirai* by[26]). Such incomplete specimens sample the known size range, and show no evidence of faintly preserved or effaced branches that would 'complete' the specimen, are not obscured by ash, and cannot be explained by general poor preservational quality. Additionally, these specimens are not aligned with other rangeomorphs inferred to have lived upright in the water column, which can be observed in close proximity but in different orientations (Supplementary Fig. 2). Importantly, at least three of these specimens show a continuation of their positive epirelief central axis beyond the broken frond margin (e.g. Fig. 2G, H, Supplementary Fig. 1D). In such cases, the midline does not continue as a zig-zag suture with the same regularity as may be expected if finer branching details had weathered away, but rather as a straight projection bending away from the specimen, which is continuous with the midline of the frond.

There is a linear relationship between specimen length and the number of first order branches in a single population of *Fractofusus* specimens from the BR5 surface (Fig. 3A, *n* = 15, Supplementary Datasets 2, 3). This contrasts with previous analyses of *F. andersoni* from the 'E' surface (*n* = 11) that found no significant relationship between specimen length and number of first order branches[13] in the explored size range (1.5–13 cm). We could not analyse individual populations of *F. andersoni* from that previous dataset because their sample includes specimens of *F. andersoni* from multiple beds, with no individual bed preserving a sufficient number of specimens to permit individual study. We note that our dataset encompasses specimens at substantially earlier developmental stages (i.e. smaller specimens) than were available for study by Narbonne and Gehling (2007)[13].

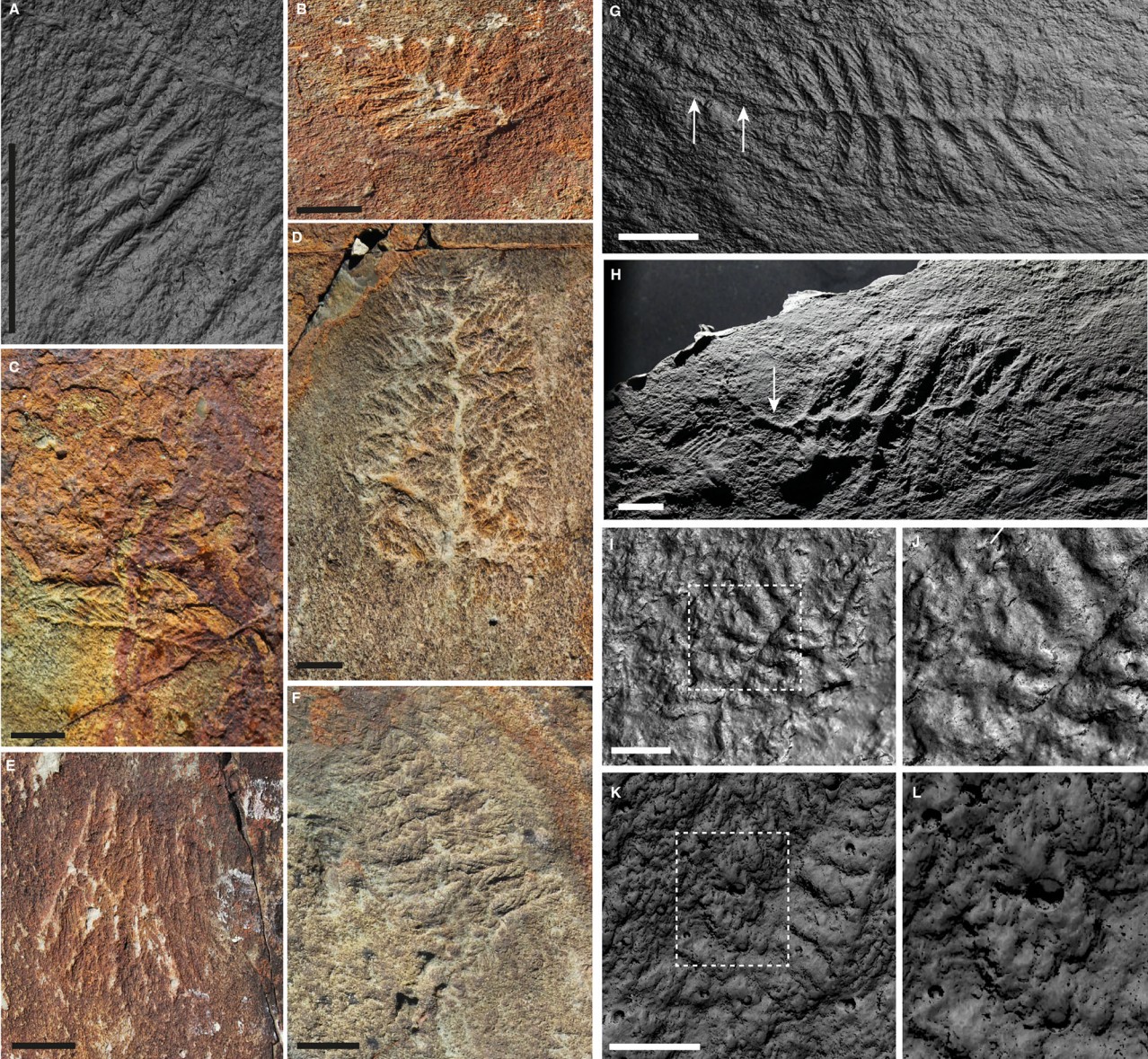

**Fig. 2 | Morphological features in *Fractofusus andersoni*. A–F** Truncated specimens with no evidence of regrowth. Specimen in **A** = X.50400.12, all others field photos from surface BR5. **G**, **H** *Fractofusus* with distal continuation of axes, interpreted as stolons. Specimen in **G** = X.50400.10 and specimen in **H** = X.50400.11. **I–L** Different orientations of first order branches, indicating they were not connected and were free to rotate. All images from specimen on X.50400.10. All individual specimen data collected from casts available as RTI files in Supplementary Dataset 1. Scale bars in **A–H** = 10 mm and in **I** and **K** = 1 mm.

In specimens where both distal tips are well preserved, there is no morphological distinction between the two distal tips (e.g. Supplementary Fig. 3B–D, G), contrary to what is observed in some other Ediacaran taxa displaying a broadly 'biterminal' shape, e.g. *Dickinsonia*[27] or *Phyllozoon*[28]. We tested for skew from a normal (i.e. symmetrical) distribution in frond outline and found that only two (length 7.8 mm and 10.4 mm) of twelve complete specimens with intercalary lengths were skewed (see Supplementary Dataset 4). Therefore, the observed skew is not systematic and we confirm that the longest pair of branches was likely centrally positioned. Moreover, in at least 14 specimens of *F. andersoni* from BR5, the longest (assumed originally central given lack of systematic skew) pair of first order branches appears to originate from a single branching point, with asymmetric offset between opposing branch pairs in the two branch vanes becoming increasingly apparent distally along the specimens (Fig. 4, Supplementary Fig. 3). The majority of specimens do not exhibit this phenomenon, with asymmetrically arranged branch pairs

evident along the full length of the specimen, as previously recorded (e.g.[13]). However, these bilaterally symmetrical specimens show no evidence of distortion, damage or overfolding around the central branches (Fig. 4, Supplementary Fig. 3), and therefore are taken to reflect a genuine biological feature.

Since the shape of the frond is not systematically skewed, measurements from the longest branch to the best-preserved distal tip are assumed to reflect the shape of entire specimens. The relationship between branch number and branch length is best described by a linear relationship for the known lifecycle of *Fractofusus* (Fig. 3D, E), though note that in a single specimen of length 34.7 mm, this relationship is best explained by a log model. The length of the longest branch increases with specimen size and is best described by a linear regression (Fig. 3B, Supplementary Dataset 5) ($p = 1.773 \times 10^{-8}$). The coefficient of these two regression lines—specimen length versus number of first order branches, and specimen length versus central branch length (Supplementary Dataset 6) – is not significantly

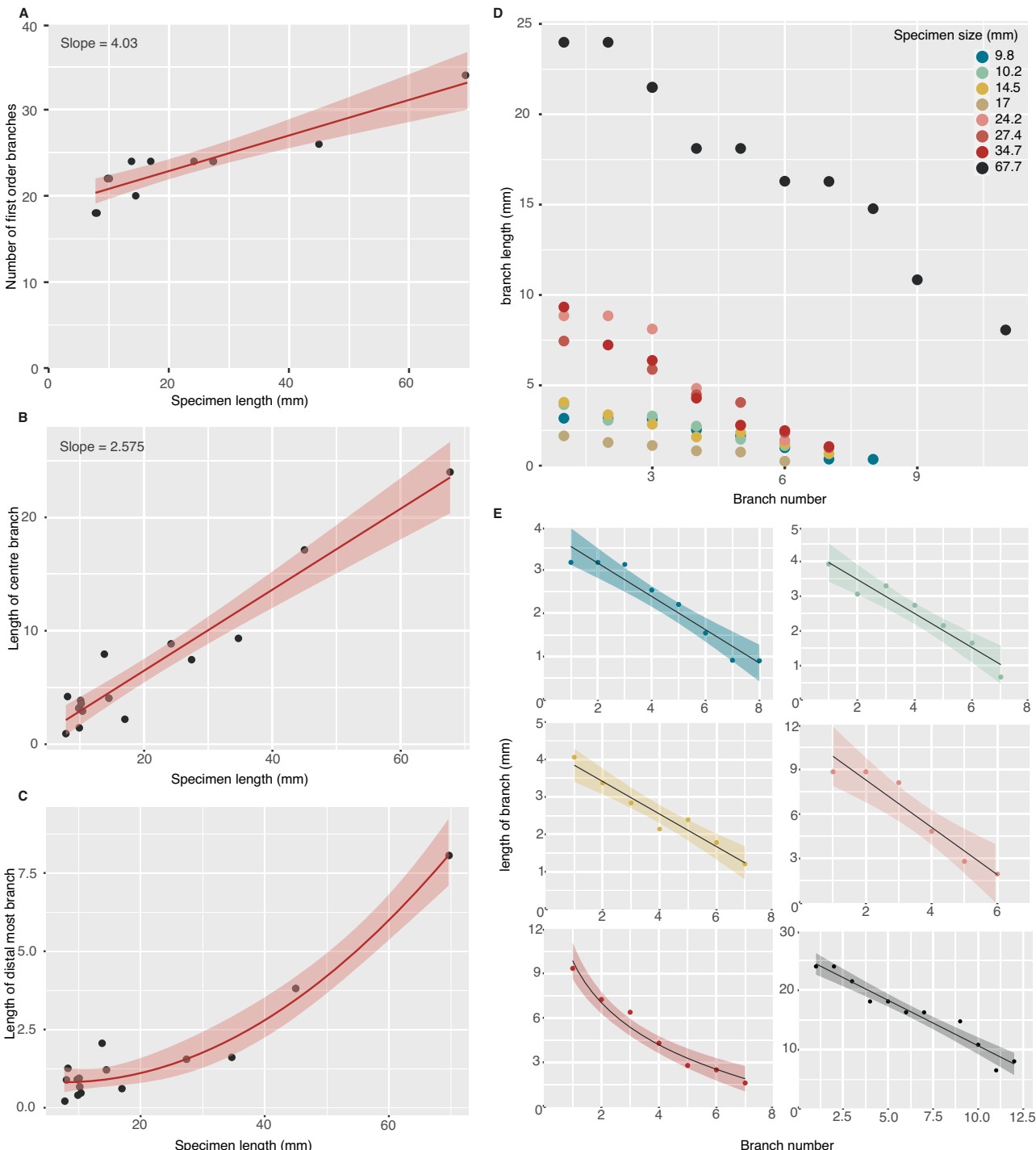

**Fig. 3 | Growth in *Fractofusus andersoni*. A** The relationship between the number of first order branches across a single row and specimen length is explained by a linear model, $p = 3.448 \times 10^{-5}$ and adjusted R squared = 0.8163. **B** The relationship between the length of the central branch and specimen length is best explained by a linear model, $p = 1.773 \times 10^{-8}$ and adjusted R squared = 0.913 **C** The relationship between specimen length and the length of the distal-most branch is best explained by a second order polynomial model, $p = 2.661 \times 10^{-8}$ and adjusted R squared = 0.9363. **D, E** The relationship between branch number (proximo-distally) and branch length,

from the central branch to the distal branch. Six representative specimens across the observed size range are shown with best fitting models in E. This relationship in all specimens bar one is best explained by linear models (p-values and adjusted R squared values: 9.8 mm = $8.915 \times 10^{-5}$ and 0.9238, 10.2 mm = 0.000434 and 0.9173, 14.5 mm = 0.0002684 and 0.9317, 24.3 mm = $1.724 \times 10^{-5}$ & 67.7 mm = $5.995 \times 10^{-7}$ and 0.9382 respectively), while a specimen of 34.7 mm is best explained by a log model: p-value of 0.9771 and adjusted R squared value of $9.396 \times 10^{-5}$. Line shows the smoothed conditional mean with 95% error bars around it.

different. The length of the distal-most (recorded) branch increases with specimen size and is best described by a second order polynomial regression (Fig. 3C, Supplementary Dataset 7) ($p = 2.661 \times 10^{-8}$).

The slope coefficient of the relationship between specimen size and branch length varies between specimens, and a model expressing

branch number against branch length given specimen size is significant, indicating that specimen size is a significant predictor of slope coefficient (0.0001706). Wald tests were used to explore this further, with specimens between 7.8 mm and 17.0 mm having statistically indistinguishable coefficients, revealing that while absolute size

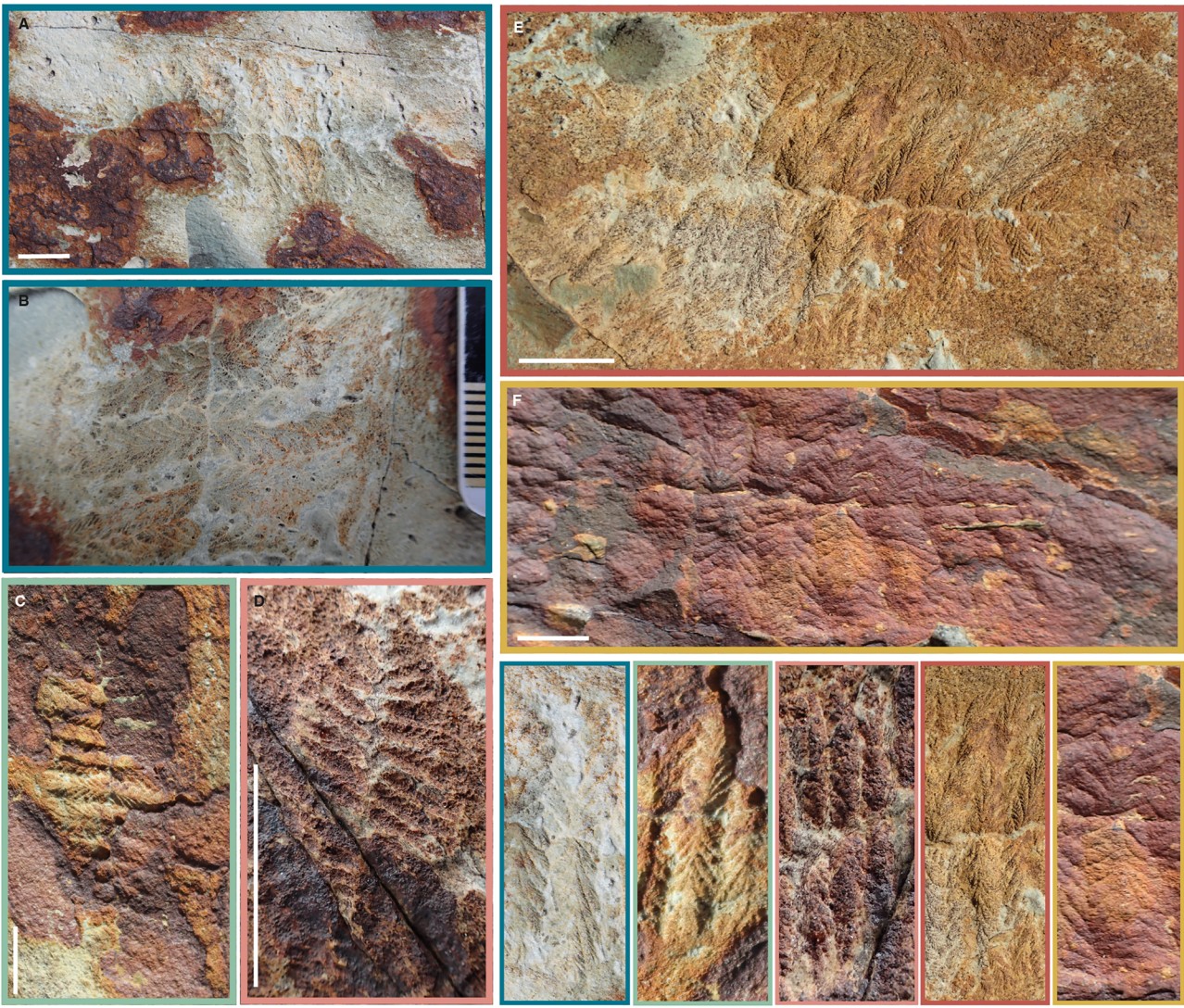

**Fig. 4 | The central branching pair in *Fractofusus andersoni*.** Field specimens of *F. andersoni* showing a bilaterally symmetrical branching pair. Opposite branch pairs are expanded at the bottom right of the figure where the first order branch alignments are magnified. The colour of the box around **A**–**F** indicates the specimen magnified in the bottom right. A&B represent the same specimen. Scale bar = 1 cm.

increased, the relationship between position and relative branch length remained constant. However, within this size class the coefficients of specimens of length 14.5 mm and 17 mm are significantly different from one another, but not from other, smaller specimens. The coefficients of specimens between 24.2 mm and 67.7 mm are statistically indistinguishable from each other but differ from all other specimens (Supplementary Fig. 4).

There is no relationship between the maximum number of second order branches on a first order branch and specimen size ($p = 0.3121$ for the best fitting log model, $n = 10$), and there is equivocal support for the number of second order branches along a given first order branch varying according to proximo-distal position in individual specimens. Only four specimens preserve sufficient second order branch information for further interrogation, of which two specimens show no significant trend (specimens of length 8.06 mm and 24.4 mm), and two did show a significant relationship (specimens of length 14.5 mm and 45 mm, described by a log model [$p = 0.009886$] and linear model [$p = 0.02024$] respectively, data available in Supplementary Datasets 2, 8).

## Discussion

Rangeomorph branches may be arranged around a central axis (monopodial) or derive directly from one another (sympodial). Our observations of a coherent, discrete midline structure indicate that branches in *Fractofusus andersoni* have a monopodial organisation, with distal branches emerging directly from a linear central axis as opposed to the central axis simply representing sediment accumulation in the gap between frondlets (proposed as a possibility by ref. 13). This central axis may extend beyond the maximum observed extent of the branches both in complete specimens (Fig. 2G; Liu & Dunn 2020[29] fig. 4A) and truncated individuals (Fig. 2H, Supplementary Fig. 3D), confirming that such specimens are not the result of folding of the organism out of the plane of preservation. Given that the central axial structure is not confined to the branching area but can continue beyond, it is best interpreted as stoloniferous[29–31].

There are multiple competing interpretations for the truncated specimens. We can rule out the possibility that they represent a deterministic developmental stage because there is no size-range bias visible in our population (Fig. 2). We can also rule out the possibility that these represent partial lifting or folding of specimens[32] because they are not consistently aligned with taxa inferred to have lived upright in the water column, which are very strongly aligned on the Brasier surface, presumably by a unidirectional current. Other potential explanations include physical damage resulting in lost frondlets, as has been suggested for visibly disturbed branches in other

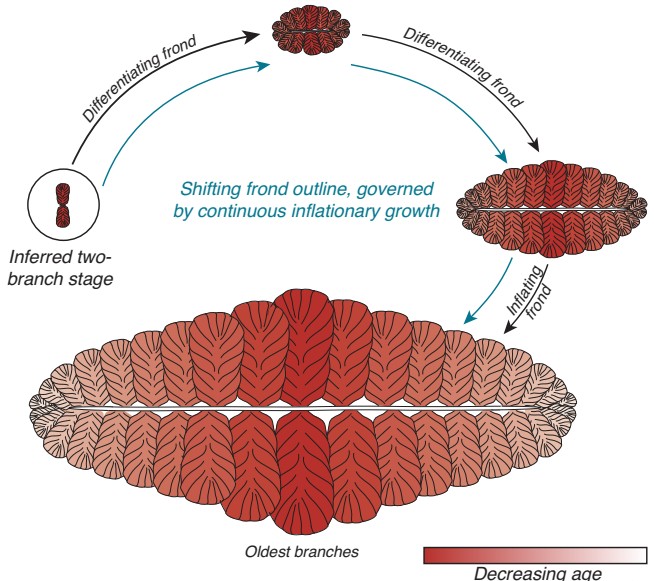

**Fig. 5 | Schematic drawing of _Fractofusus andersoni_ morphogenesis.** Shift in developmental mode (from differentiation-heavy to inflation-heavy) is shown with black arrows and annotation, and shift in frond-outline is shown with blue arrow and annotation. Oldest branches are shown in red, with progressively younger branches getting lighter, reaching white in the largest specimen with the most branches. Colours of branches in all other specimens correspond to the largest specimen and branches of the same colour are homologous.

rangeomorph taxa[33], or reproductive strategies including budding or fragmentation, if reproductive maturity had been reached by our truncated specimens. We are currently unable to discriminate among these interpretations.

Where preservational fidelity allows reconstruction of the branching arrangement, the majority of specimens show an offset longest branching pair, but at least 14 show central branches that lie directly opposite one another. An opposite arrangement could result from rotational variation or post-mortem deformation, but this would be difficult to achieve and an offset arrangement would be more likely to result from such processes. Therefore, two possibilities exist: (1) an opposite arrangement was the original branching pattern for the central branches, representing the oldest branches from which the central branching axis diverged laterally, or (2) the arrangement of first order branches was not tightly controlled during morphogenesis, producing specimens with both offset and opposite morphologies.

Our data support the conclusions of previous authors[8,13] that _Fractofusus_ differentiated first order branches from both distal tips. An expectation of such a bipolar growth model is that growth proceeds equally from both tips: if growth proceeded from one tip only, we would expect frond asymmetry, as the position of the longest branch would move relative to the centre of the specimen and this would likely correspond to overall size (e.g. ref. 34). We find no evidence for systematic skew in the frond outline, and therefore conclude that _F. andersoni_ was originally symmetrical and bipolar. Additionally, we observe many specimens of _F.andersoni_ 'broken' along the principal axis, without any evidence of regrowth (Fig. 2A–H). None of those specimens tend towards ivesheadiomorphs (fossils interpreted by some as the decaying remains of rangeomorph fronds[35]), implying that they were unlikely to have represented degraded necromass at the time of burial (though they could represent specimens damaged shortly prior to burial). These data may suggest that _Fractofusus_ only retained the ability to generate new first order branches at the distal tips, and that damage to a particular tip led to the cessation of growth from that point. Taylor et al.[26] have suggested that in _Fractofusus_

_misrai_, first order branches could differentiate anywhere along the monopodial axis, serving to redirect the principal axes of the growing frond. We find no evidence in our data to support this interpretation for _F. andersoni_.

_Fractofusus andersoni_ differentiated first order branches sequentially over time, and each first order branch continued to inflate during growth (Fig. 3). The pattern of inflationary growth changed across the life cycle, as recorded in changes to the slope coefficient relating branch number to branch length (Supplementary Fig. 4). This indicates a gradual shift in frond outline, with less mature specimens having a more linear outline, and more mature specimens having a diamond-shaped outline (Fig. 3D). We consider these transitions to be developmentally regulated and not a response to environmental variation, given that developmental changes are likely to be continuous and therefore the shape changes can be explained by one gradual shift. If the changes were a response to environmental change, this would require three discrete shifts/changes from naturally equal slope coefficients. The oldest, central branches enlarge approximately isometrically with branch differentiation, although we note large error in the relationship between number of first order branches and specimen length (Fig. 3A). As such, the shift in inflationary profiles is likely driven by increasing developmental time between the differentiation of central and distal branches, which individually grew at a consistent rate in our examined size range.

The second order polynomial relationship between specimen length and the length of the distal-most branches indicates that the relative rate of branch differentiation slows across the known growth cycle. The increasing size of the distal-most branches reflects successively greater periods of time before the differentiation of another branch, but we do not observe a size at which the differentiation of new branches ceases. It is possible that _Fractofusus_ went through a differentiation-dominant phase of growth very early in its developmental sequence[10], with the smallest reported _Fractofusus_ in this study being 7.8 mm in length and possessing 18 first order branches in total. Our interpretation of opposite branches as a likely original biological feature may suggest that _Fractofusus_ began the frondose stage of its life cycle with just two, opposing, first order branches. An alternative scenario could see a series of branching pairs differentiating simultaneously, but that provides no explanation for the presence of two, opposing first order branches at the centre of the organism as opposed to any other position. Therefore, we prefer the former hypothesis based on our data but note that we cannot rule out the latter.

There is no obvious change in the number of second order branches per first order branch along the proximo-distal axis. This suggests that second order branches do not differentiate continually during the development of individual first order branches, but testing this would require additional data not available to us. It may also be that the distal—presumed least mature—branches do possess fewer second order branches, but we are unable to consistently identify this signal because differentiation is not the principal developmental mode during the measured part of the life cycle (Fig. 3), and the details of first order branches at the distal apices of the organism are typically the least well-preserved portions of the specimens.

Our morphogenetic model (summarised in Fig. 5) shows the sequential differentiation of branches across the examined life cycle in _Fractofusus_, with a transition away from differentiation-heavy growth in the largest specimens. The relationship between inflationary growth in first order branches and specimen size changes, becoming more prominent when compared to the differentiation of branches and resulting in a gradual shift in the outline of the frond. Our finding of a single pair of bilaterally arranged branches in multiple specimens of _F. andersoni_ is compatible with the earliest frondose phase in the _Fractofusus_ life cycle being composed of two opposing branches, from which subsequent branches emerged distally and from a central stolon. However, a competing hypothesis is that _Fractofusus_ shows

developmental plasticity in the arrangement of branches around the central axis. We find this less likely, given the bilateral branching pair are only ever found in the centre of the specimens, which would not be an expectation of this model. Future data from other fossil surfaces will confirm or refute this hypothesis.

In *Charnia masoni*, a uniterminal rangeomorph, the initial primary growth mode is the differentiation of branches, ultimately becoming secondary to the inflation of pre-existing branches[10], as with *Fractofusus*. However, these patterns manifest at different specimen sizes in these taxa, with the differentiation of branches being the principal developmental mode in *Charnia* up to specimen sizes of minimally 22 cm (other taxa, presumed closely related to *Charnia*, show similar profiles: e.g. *Trepassia*[36]). The multifoliate rangeomorph *Bradgatia* exhibits individual fronds arranged around a central, small holdfast disc from which branches emerge (Fig. 1B[37,38]), and which is often inferred to be more closely related to *Fractofusus* than *Charnia* (e.g ref. [12]). *Bradgatia* shows the differentiation of first order branches only early in morphogenesis[37], as in *Fractofusus*. The morphogenetic pattern seen in *Fractofusus* and *Bradgatia* is less similar to what is known of *Pectinifrons* (Fig. 1D), another taxon that has been suggested to be closely related to *Fractofusus*[12]. Bamforth et al.[39] report the differentiation of branches in *Pectinifrons* as continuing across the known size range, however those authors consider specimens from multiple populations/sites in their analysis, potentially introducing confounding ecophenotypic variation.

Dunn et al.[10] proposed a scheme to find homology between disparate rangeomorph taxa and suggested that the entire frond of *Charnia* was homologous to a single first order branch of *Fractofusus*. However, that study did not go further in attempting to find homology between the non-frondose components of rangeomorph bodyplans and so it was not possible to rationalise the disparate rangeomorph bodyplans. Our novel data on *Fractofusus* help resolve this: recent work has shown that in at least some uniterminal fronds, potentially including *Charnia*, individuals of the same species and of significant size (indicating a role extending beyond simple reproduction) had filamentous connections[29,31], similar in expression to the central stolon of *Fractofusus* that we present here. This observation provides an obvious way to rationalise between the bodyplans of these rangeomorphs, with differences in gross morphology explained by simple compression and/or extension of the fundamental axis (that from which the lowest [thus largest in size] order of branches derive), with individual fronds differentiating distally along a stoloniferous extension. This model is easily applicable to most other rangeomorph taxa, except bush-like forms including *Bradgatia*. There, the fundamental axis is cryptic but it must represent the axis along which branches differentiate from the holdfast. Following Dunn et al.[10] each individual folium in *Bradgatia* is homologous with the entire *Charnia* frond, or a single first order branch in *Fractofusus* (as it contitutes the largest branching unit that is independent of all others). A mechanism by which to generate, or lose, a bush-like morphology is therefore suggested to be contraction or extension of a lateral branching axis.

This model equates single first order branches in *Fractofusus* with multiple, interconnected uniterminal fronds that have been interpreted as clonal or colonial[29] and therefore could suggest that *Fractofusus* itself may be a clonal colony. If this is the case, *Fractofusus* would show extremely high levels of colony integration, with a bodyplan that remained static across its multi-million year stratigraphic range[6]. This model is not incompatible with our morphogenetic data which show that the position of the branches along the distal-proximal axis scales linearly with length in the examined size range, unlike in *Charnia* where there is a second order polynomial relationship between branch length and position in the apical-basal axis[10] indicating tighter spatial regulation of branching growth than in *Fractofusus*.

Branching forms are ubiquitous in nature and many different animal groups have achieved a form of branching morphogenesis,

spanning a variety of scales. Whole animal branching morphogenesis is almost exclusively the result of the evolution of a colonial, modular form (e.g. corals, bryozoans) from a unitary ancestor[40]. Branching growth in such arborescent animals (following Van Valen 1978[41]) is largely non-deterministic, and there is often a high degree of morphological plasticity in response to environmental parameters[42,43]. Rangeomorph growth, by contrast, is highly deterministic, but nevertheless compelling data show some taxa are truly modular[33]. In both *Fractofusus* and *Charnia*, branching is stereotypic at all known orders and is tightly controlled, despite hundreds of specimens being known, in the case of *Charnia* from many different fossil localities (e.g ref. [10,44–49]), implying that their core morphogenetic programmes were refractory to variation in environmental conditions. However, whereas morphogenetic pattern at the level of the bodyplan appears to be different between rangeomorphs and other arborescent animals, there are possible similarities in branching mechanism. Monopodial hydroids maintain their stem by inhibitory signals from two laterally growing polyps; the stem terminus is itself a polyp primordium[50]. This may provide an analogous mechanism for the differentiation of branches in multifoliate rangeomorph taxa, whereby the inhibitory signal is removed at different stages, and the terminal (distal-most) branch forms.

Recent work has suggested that branching in eukaryotes may follow common patterns of morphogenetic regulation, whereby tips are equipotent and branch stochastically—and non-deterministically—to fill space, terminating once in close-proximity to neighbours[51]. Other data suggest a more complex picture, with branching in the mammalian kidney being deterministic[52,53] and therefore not conforming to a stochastic branching and annihilating random walk model, while still being governed by tip growth and neighbour repression[53]. On this basis, some researchers have proposed general rules for branching morphogenesis, where eukaryotic branching is generally explainable as the interaction of branch tips with their local environment[53]. Coudert et al.[53] further suggest that inhibitory cues may dominate over branch-promoting factors given that hyperbranching phenotypes (associated with a loss of inhibitory cues) are more common than hypobranching phenotypes in the ascomycete *Neurospora crassa*. Rangeomorphs retained the latent capacity to generate new branching axes, but this is repressed in most circumstances[33], providing some support for this hypothesis. The branching morphogenesis of rangeomorphs conforms to what is known in living eukaryotes, but goes beyond what living (and other known fossil) eukaryotes achieve in that the core branching pattern is not impacted by environmental variation (insofar as we have examined a time-averaged population of *F. andersoni*, see ref. [48] also; we note that there is limited data available for some colonial animals, e.g. sea pens).

Rangeomorph bodyplans were not defined by environmental parameters (*contra*[25,54]) and do not appear particularly malleable[10,48]. Genus level morphological differences cannot be explained via heterochrony[54] and discrete character changes preclude ecophenotypism. The developmental biology of rangeomorphs therefore supports suggestions that their morphological evolution is best explained by adaptation in distinct, diverging lineages[54]. Furthermore, our results indicate that rangeomorph development was sufficiently tightly regulated that it is possible to derive morphogenetic principles that underpin variation in the whole clade. Much recent work has looked at the influence of the evolving regulatory genome on the evolution of animal complexity[15,55,56], but our morphogenetic data suggest that early animal lineages, which lived and died long before the Cambrian, display the hallmarks of a tightly regulated morphogenesis. Traditional terminology describing growth in rangeomorphs as a consequence of 'inflation' or 'insertion'[8] implies that the distinctive reiterative and seriated modular bodyplans of rangeomorphs required less regulation overall than those of crown-eumetazoans. However, there is evidence to suggest that individual fronds in certain rangeomorph taxa exhibit

modular development[33], which would require complex underpinning developmental regulation. Valentine[57] argued for such a scenario in rangeomorphs and other Ediacaran macrofossils, highlighting that compartmentalisation of gene-expression (required for a modular bodyplan) is commonly achieved through cis-regulation of early-acting genes implicated in development.

Further support for this scenario comes through analysis of developmental abnormalities, which provide information about the constraints enforced by developmental regulation as growth proceeds normally. The identification of abnormalities therefore provide an independent test of our model of morphogenesis, which is based on specimens that have developed normally. Such abnormalities have been reported (though rarely) in both rangeomorphs and dickinsoniomorphs, confirming that morphogenesis of both groups must be driven by modular regulation. This is evidenced in rangeomorphs by the abnormal anatomical reversion to a higher branching order[33], and in *Dickinsonia* by the abnormal specification of additional main body axes[58]. Following Valentine (2001)[57], this does not support hypotheses suggesting minimal regulation of development in these organisms.

Living eumetazoans are characterised by highly complex spatially and temporally controlled organ systems, including guts, muscles and nervous systems. Rangeomorphs appear to have diverged before the appearance of some or all of these key synapomorphies, but nonetheless display differentiated tissues and perhaps organs. *Fractofusus* is unlike most other rangeomorphs in that it is constructed entirely of frond branches and a stoloniferous central axis, whereas other taxa possess structures including holdfast discs, stems and stalks which differ in their construction to the frond itself. The functional morphology of rangeomorphs remains controversial[59,60] but the morphological distinction between the frond, holdfast, stalk and stem is compatible with their interpretation as organs, in the sense that these represent differentiated anatomical units specialised to perform specific functions. Therefore, the fundamental regulatory pathways that establish body regionalisation must have already been established and co-opted into these roles deep within the eumetazoan stem-lineage, pathways that were elaborated in the construction of crown-group eumetazoan organ systems.

Morphological differences between otherwise disparate rangeomorph taxa can be rationalised by study of underlying developmental pattern and morphogenetic principles, providing a mechanism to explain much variation and innovation in rangeomorphs. We conclude that rangeomorph growth was conserved and predictable, and our data imply a morphogenetic strategy that was highly regulated, demonstrating that the most ancient eumetazoan fossils known already manifest evidence of complex developmental regulation. Ediacaran macrofossils must be considered in hypotheses concerning the evolution of developmental regulation in metazoans, and novel developmental data from additional members of the Ediacaran macrobiota – including developmental abnormalities – will continue to refine our picture of early animal evolutionary history and the mechanisms that underpinned the diversification of animal life.

## Methods

We acquired permissions from the Mistaken Point Ecological Reserve (MPER) and the Parks and Natural Areas Division of the Government of Newfoundland and Labrador before conducting scientific research within MPER including access. Fossil surfaces within the reserve are accessible only via permit and fall within the Mistaken Point UNESCO World Heritage Site - contact the reserve manager for more information.

More than 100 specimens of *Fractofusus andersoni* from the 'Brasier Surface' (Briscal Formation, Bed BR5; see ref. 61) were studied either in the field or from high resolution jesmonite casts and silicon rubber moulds reposited at the Sedgwick Museum of Earth Sciences (University of Cambridge, U.K; CAMSM X.50400.1 – 50400.17). The Brasier Surface has been dated to 567.63 ± 0.66 Ma[6].

Specimens were retrodeformed prior to quantitative study to account for tectonic deformation using the constant area method (e.g.[62]). A subset of 18 specimens (ranging from 7.8–69.7 mm in length) were used in analyses of growth, selected on the basis of their quality and completeness of preservation (comprising specimens that exhibit at least one apparently complete distal tip and complete branches across the rest of the body), sampling the full range of observed specimen sizes with complete representatives on the surface (Supplementary Datasets 2–8, specimen list of all accessioned material present in Supplementary Dataset 9). Fossilised populations of *Fractofusus* from other horizons were not considered due to the potential confounding impact of ecophenotypic and taphonomic variation, given that individual surfaces all likely experienced different environmental conditions[63]. The *Fractofusus* population examined here may not represent a population in a biological sense because the preserved community has likely undergone time-averaging, as has been documented previously for other Avalonian Ediacaran fossil surfaces[35,47].

Total specimen length, number and length of first order branches, and number of second order branches—from which first order branches are constructed – were documented for each individual specimen (data available as Supplementary Dataset 1). Smaller (higher) branching orders were not assessed, since preservational differences between specimens hindered comparison. Different models of growth and development were compared using AICc values to account for small sample size, and then assessed for significance (following[10]). Branch number was plotted against branch length across entire specimens. Data (here representing frond outline) skewness was assessed using the D'Agostino test and the coefficients of regression models were compared using Wald tests. All analyses were performed in R[64] and code is available in Supplementary Code 1.

Two species of *Fractofusus* are known, but we only consider growth data from a single population of *F. andersoni*, at a site where no obvious specimens of *Fractofusus misrai* have been observed amongst the population of >100 *Fractofusus* specimens. Differences in branch organisation in more mature specimens[13] render uncertain points of branch homology between the two species. There is also uncertainty about whether the two species of *Fractofusus* are morphologically distinct at their earliest developmental stages. As such, we cannot at present compare quantitative branch data derived from these two species.

### Reporting summary

Further information on research design is available in the Nature Portfolio Reporting Summary linked to this article.

## Data availability

All quantitative data and image data not drawn from a single photograph are available in Supplementary Datasets 1–9. Source data for Fig. 3 can be found in Supplementary Datasets 2 (panels D&E), 3 (panel A), 4 (panel B) and 8 (panel C). All data required to reproduce the results presented in this manuscript are provided as supplementary data and all source data for graphs are presented in Supplementary Datasets 2–8. We have provided a full list of specimens used in this study, along with their institutional accession numbers. Specimens are deposited at the Sedgwick Museum of Earth Sciences and under the care of Matt Riley: mlr44@cam.ac.uk.

## Code availability

The R script used to run all analyses is available as Supplementary Code 1.

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

## Acknowledgements

We acknowledge J. Matthews and S. McMahon for assistance in the field, and A. Dennis for assistance with casting. The authors acknowledge support from the Natural Environment Research Council (NE/L002434/1, NE/V010859/2 & NE/W00786X/1 to F.S.D; NE/P013678/1 to P.C.J.D. and A.G.L.; and NE/L011409/1 to A.G.L.), Biotechnology and Biological Sciences Research Council (BB/N000919/1; BB/Y003624/1 to P.C.J.D.), Leverhulme Trust (RF-2022-167 to P.C.J.D. and RPG-2021-052 to A.G.L.), the Royal Commission for the Exhibition of 1851 (to F.S.D), and the John Templeton Foundation (JTF 62220; JTF 62574 to P.C.J.D.). We thank two anonymous reviewers who made this manuscript substantially stronger.

## Author contributions

FSD, PCJD and AGL conceptualised the project. FSD and AGL collected all data. FSD analysed and interpreted the data, made the figures and wrote the first draft of the manuscript, with input from all other authors. All authors approved the final version of the manuscript.

## Competing interests

The authors declare no competing interests.
