## [Transparent Peer Review file · Nature Communications]

Morphogenesis of *Fractofusus andersoni* and the nature of early animal development

Corresponding Author: Dr Frances Dunn

Version 0:

Reviewer comments:

Reviewer #1

(Remarks to the Author)

The manuscript by Dunn and colleagues describes the growth of the Ediacaran rangeomorph *Fractofusus andersoni* based on data collected from a population of more than 100 individuals. I find that the authors have provided an excellent, thorough investigation detailing the growth of this taxon. The implications of highly regulated development are well-founded. I believe this manuscript will be of interest to anyone curious about the growth dynamics of early eumetazoans and will be suitable for publication pending revisions suggested below.

Line 46: Reference for the suggestion that morphological variability is the result of selective pressure.

Lines 46-48: These are interesting end-member interpretations, but I am not sure they apply to the study? By limiting your investigation to one species, you are assuming morphological variation producing diverging lineages and that different morphologies do not reflect life histories. The authors also acknowledge that because they limit their study to one population, they cannot address ecophenotypic variation. I would suggest this paragraph be rewritten to establish the hypotheses being tested regarding the developmental capacity of rangeomorphs and how those might apply more broadly to the early evolutionary history of animals.

Line 61: Can the authors elaborate on why this constitutes a “new population”? I am unfamiliar – but clearly this surface was dated in a publication from 2021 (Matthews et al) and the authors provide reference to a publication from 2016 (Liu et al). Is it that specimens of *Fractofusus* have not been described in detail, or is this an entirely new surfaces discovered by the authors?

Lines 77-78: Can the authors comment at all on how retrodeformation might impact their results. I am confident this is a valid methodology, but am simply curious whether there is any potential influence of this process in the models produced.

Lines 78-80: Can the authors clarify – are these the only 18 specimens for which one distal tip is complete, or were they selected for some other reason.

Line 81-84: Throughout the manuscript ecophenotypic variation is discussed, but as stated here cannot be evaluated because the authors only assess a single population. I would suggest the implications of the manuscript would be much more robust (as stated in the introduction) if they could assess the possibility of environmental controls on morphology. A simple test would be to plot any number of specimens from other bedding horizons/localities on some/all of the graphs produced in figure 3 and see if the variation in these specimens falls outside or within that observed. PCA or similar multivariate analysis of morphological variation within this population compared to that of all *Fractofusus andersoni* specimens available could also be employed. Even if a comparable number of specimens are not available from a single ‘other’ population (as expressed in line 122-124), the authors can test the hypothesis that including specimens from any other environments/beds/localities should introduce greater variation, providing at least some opportunity to evaluate the possibility of ecophenotypic controls on morphology. This would greatly increase the significance of the findings. Alternatively, if the authors are uncomfortable or cannot do this, they should eliminate all text in the discussion that references ecophenotypic variation (e.g. line 328-330, and all conclusions that follow) as they have not evaluated this factor by their own admission.

Line 98-100: Is this an original hypothesis of these authors (that there is little or no difference between the species in the earliest developmental stages)? Or can they reference other work? Do the results found here offer any new insight? This is seemingly a major contribution of such a detailed study.

Lines 109-115 and 183-187: These incomplete specimens are intriguing, and more description is warranted here. First, the authors have not confirmed that these are not the result of folding out of the plane of preservation as stated in lines 185-186. This has been observed in other Ediacara taxa and can be tested by examining whether the incomplete portion in different specimens displays any alignment that can be attributed to current (cf. Evans et al., 2015 P3). While I agree specimens with a midline that extends into these otherwise incomplete sections are unlikely the result of folding – an alternative is that the branches on both sides of this midline are each folded. This may be expected if the midline is a point of attachment and/or more rigid than branches, and the fact that it only occurs in 3 individuals is consistent with inferred low probability. I fully expect alignment to be random, which would then falsify the folded/lifted hypotheses, but this data should be presented.

The authors should be more explicit about their hypothesis for why these specimens are incomplete. They rule out taphonomic explanations, but what caused this? Did development cease in one direction due to some environmental or size factor? Or is this a 'normal' pattern of growth to be expected in all complete specimens? Further elaboration is warranted, even understanding that definitive conclusions may not be possible. Further justification for the stoloniferous interpretation is necessary. Is this the only other non-taphonomic hypothesis that can explain these incomplete specimens?

Line 191-195: An alternative explanation for the observed patterns is that the arrangement of first order branches was not tightly controlled during the development of *Fractofusus*, producing specimens with both offset and opposite morphologies. This may be a significant result, suggesting some amount of developmental plasticity in early eumetazoan patterning around a central axis. Discussion of this possibility and any observations to support or refute (e.g., does retro deformation tend to minimize or exaggerate offset?) this hypothesis is warranted.

Line 204: Can you provide a reference to figured specimens that exhibit this 'broken' morphology?

Lines 205-210: What if such damage occurred relatively close to the time of burial?

Line 213-215: This statement could lead to the assumption that no supporting data was published by Taylor et al., 2022 – however, there is supplementary data available (although it does not seem to include the metrics needed to address the hypothesis discussed here). I suggest simply stating that "We find no such evidence in our data to support this interpretation for *F. andersoni*."

Line 237-242: Why is the hypothesis of 2 opposing first order branches supported over any other number – certainly the y-intercept of figure 3A would be much greater than 2 – suggesting many more initial branches? Additionally, the lack of consistent labelling in figure 3 leads to some confusion. Please add more explicit indication that the y-axis in figure 3A does not begin at 0. Why is zero labelled in the y-axis of some panels and not others, or for the x-axis in any panel? Convention suggests that the origin of each graph begin with (0,0) and be labelled accordingly.

Morphogenesis: Can the authors provide a figure that documents changes in *F. andersoni* through the developmental stages observed. Something like Figure 4 of Dunn et al., 2021 *Science advances*?

Line 261-289: The arguments of this section are difficult to follow. This study details a single population of a single species, the conclusions drawn here between other rangeomorphs are therefore incomplete and poorly defined. Much of the argument seems to simply state that different fronds have different morphology. Figure 5 does not add to this, displaying 3 different fronds but failing to "rationalize variation between them" (as stated in the abstract) given the arrows suggest changes could proceed in any direction? This figure may be redundant since various rangeomorph morphologies are displayed in Figure 1. The model presented seems to hinge around changes to the main axis, but in one of three key taxa discussed (*Bradgatia*) this is cryptic and so must be inferred. If morphology of the main body changes it follows that the main axis changes too, what evidence is there that this is the controlling factor? This section is likely better suited to a separate manuscript dedicated to fully elaborating these changes and including observations from a significant proportion of Rangeomorph taxa, rather than the few examples selected and the incomplete discussion that follows.

Line 281: Lowest order (thus largest in size)?

Line 293-295: Provide references to this claim. Is this true of modular taxa, or just colonial forms? Would you consider *Fractofusus* colonial?

Line 355: Double negative, consider revising.

Line 366-365: This statement (and similar statements elsewhere – e.g., lines 28-29 of the abstract) is not supported by the main body of the text. Animals today exhibit variable "developmental regulatory capacity" (e.g. number of Hox genes), where does *Fractofusus* fit within this range? More importantly, this statement oversimplifies the distinction between possessing the gene regulatory tools necessary to produce complex morphologies (which likely extends well beyond metazoan lineages) and the co-option of those tools that results in their expression. The current manuscript documents that growth in *Fractofusus* was modular and highly regulated – an important find and one worth publishing. However, this is certainly less complex than the regulatory pathways that need to be established to produce differentiated body regions, organs, appendages, etc. found later in the fossil record.

(Remarks on code availability)

Reviewer #2

(Remarks to the Author)

In this manuscript, the authors examine growth and morphogenesis in the Ediacaran taxon *Fractofusus* on a single bedding plane. *Fractofusus* is one of the more iconic and generally well-known taxa of the suite of Ediacaran fossils representing the oldest of the Ediacaran fossils – commonly referred to as the Avalon assemblage. *Fractofusus* is part of a group of taxa that exhibit self-similar branching that are broadly referred to as rangeomorphs sharing a similar branching mode and these morphotypes characterize these oldest Ediacaran assemblages.

The authors demonstrate effectively that *Fractofusus* had highly regulated growth. This is a new and important contribution. Detailed studies of growth of the enigmatic Ediacara biota are few but notably include the similarly enigmatic *Dickinsonia*. While there have been a number of studies of growth in the rangeomorphs – more than any other Ediacaran “clade”, this study represents a new approach and insight. A single bedding plane eliminates a number of potential environmental and taphonomic issues. Furthermore, the authors address morphogenesis in a novel but detailed manner.

The results are clear cut and I think very straight forward. While this does not radically change our interpretation of *Fractofusus* or rangeomorphs, it clearly shows that we need to regard them from a very different perspective than just that they are “weird and different” and that there is no way to constrain them on the tree of life. That’s important not just for these taxa but for understanding the early evolution and evolutionary experimentation among early forms. The documentation of highly regulated growth amongst Earth’s earliest multicellular heterotrophs is important.

The authors push the importance of teratologies (line 348 onwards) which actually detracts from the strength of the paper. At best, these are rare. Many would argue with the *Dickinsonia* interpretation and the authors do not cite a rangeomorph example. This feels like a strawman and is not necessary. They have adequately demonstrated highly regulated growth – teratology is not key and pulls the argument down.

I also don’t think that the authors need to emphasize the Precambrian-Cambrian boundary. I know that in some circles, this is a “put your finger on it” evolutionary moment. But nowadays most don’t see it that way. I think a better way to approach would be to put this all in the context of early animal evolution not tying it to a stratigraphic boundary that is really just about the appearance of one trace fossil. The main part of the Cambrian radiation was much later – but a great deal of evolution happens between the Avalon assemblage and the Cambrian radiation. Putting this in a much broader context would be useful to readers from a broader audience.

Line 42 – I wouldn’t say striking given the diversity of morphology in animals – in the end, the variation is impressive given that they are much of a muchness – that is all grow in a self-similar way.

Line 54 – typo – *Dickinsonia* not *Dickinsonia*

125 – assume that this means that they were just smaller?

329 – Highly regulated growth does not mean that they were not malleable. It is hard to really say without comparisons to populations on other beds. There are certainly organisms with highly regulated growth that are indeed malleable.

(Remarks on code availability)

Version 1:

Reviewer comments:

Reviewer #1

(Remarks to the Author)

The authors have addressed all of my previous concerns. I find the manuscript sufficient for publication and believe it will be of broad interest to the readers of Nature Communications.

(Remarks on code availability)

Our responses are shown below in blue.

Reviewer #1 (Remarks to the Author):

The manuscript by Dunn and colleagues describes the growth of the Ediacaran rangeomorph *Fractofusus andersoni* based on data collected from a population of more than 100 individuals. I find that the authors have provided an excellent, thorough investigation detailing the growth of this taxon. The implications of highly regulated development are well-founded. I believe this manuscript will be of interest to anyone curious about the growth dynamics of early eumetazoans and will be suitable for publication pending revisions suggested below.

We would like to thank the reviewer for their thoughtful comments and suggestions, which have substantially improved our manuscript.

Line 46: Reference for the suggestion that morphological variability is the result of selective pressure.

This is the Brasier and Antcliffe 2004 paper which is referenced later in the sentence, but we have now added a reference after this clause for clarity.

Lines 46-48: These are interesting end-member interpretations, but I am not sure they apply to the study? By limiting your investigation to one species, you are assuming morphological variation producing diverging lineages and that different morphologies do not reflect life histories. The authors also acknowledge that because they limit their study to one population, they cannot address ecophenotypic variation. I would suggest this paragraph be rewritten to establish the hypotheses being tested regarding the developmental capacity of rangeomorphs and how those might apply more broadly to the early evolutionary history of animals.

The reviewer is right and we have implemented this suggestion in full. The relevant text now reads:

“Fractofusus thus represents an ideal model system in which to establish morphogenetic principles from which the development of other rangeomorphs may be rationalised, providing a framework for discriminating among competing hypotheses for the generation of rangeomorph form and developmental evolution more broadly. This is significant because previous studies have suggested

that step-changes in developmental regulation presage the diversification of animal life during the early Cambrian Period (e.g. ¹⁴). The ichthyosporean, filasterean and choanoflagellate relatives of animals possess orthologues of genes implicated in the development of animal traits (e.g. multicellularity^{15,16}), but have not evolved beyond the intercalation of a multicellular stage in their otherwise unicellular life cycles¹⁷. An often-cited reason for this apparent limitation is the simplicity of their underlying developmental regulation, which plays out temporally, rather than temporally and spatially as in animals^{18,19} (though see²⁰). The evolution of developmental regulation is therefore widely implicated in a number of key transitions, including correlations between the origin of animals and distal regulation^{15,21} and the diversification of bilaterians and distal enhancer elements²². Understanding such step changes requires knowledge of developmental pattern in animal groups which subtend major nodes in the animal tree. For Eumetazoa, such knowledge can be obtained by understanding the morphogenetic basis of distinct rangeomorph morphologies.”

We have kept reference to these original hypotheses in one paragraph in the discussion, but have rewritten the text so as to be clearer that discussion of ecophenotypism in rangeomorph bodyplans does not come from our study:

“Rangeomorph bodyplans were not defined by environmental parameters (contra ^{24,57}) and do not appear particularly malleable^{9,51}. Genus level morphological differences cannot be explained via heterochrony⁵⁷ and discrete character changes preclude ecophenotypism. The developmental biology of rangeomorphs therefore supports suggestions that their morphological evolution is best explained by adaptation in distinct, diverging lineages⁵⁷.”

Line 61: Can the authors elaborate on why this constitutes a “new population”? I am unfamiliar – but clearly this surface was dated in a publication from 2021 (Matthews et al) and the authors provide reference to a publication from 2016 (Liu et al). Is it that specimens of *Fractofusus* have not been described in detail, or is this an entirely new surfaces discovered by the authors?

We apologise for the confusion, we have removed reference to ‘new’ population. The reviewer is correct that the surface the fossils occur on has been known for some time, but the organisms on it have not been described in detail until now.

Lines 77-78: Can the authors comment at all on how retrodeformation might impact their results. I am confident this is a valid methodology, but am simply curious whether there is any potential influence of this process in the models produced.

The impact of retrodeformation on *Fractofusus* would likely manifest in stochastic length-width ratios to specimen size because *Fractofusus* specimens are not oriented in a single direction. Given we find clear and significant relationships in our data, we think there is unlikely to be significant influence of retrodeformation on our measurements.

Lines 78-80: Can the authors clarify – are these the only 18 specimens for which one distal tip is complete, or were they selected for some other reason.

These were the only 18 specimens where we could be sure that at least one distal tip was either complete or very substantially complete, and that were preserved along their entire length. We have now clarified this in the text: *“(comprising specimens that exhibit at least one apparently complete distal tip and complete branches across the rest of the body)”*.

Line 81-84: Throughout the manuscript ecophenotypic variation is discussed, but as stated here cannot be evaluated because the authors only assess a single population. I would suggest the implications of the manuscript would be much more robust (as stated in the introduction) if they could assess the possibility of environmental controls on morphology. A simple test would be to plot any number of specimens from other bedding horizons/localities on some/all of the graphs produced in figure 3 and see if the variation in these specimens falls outside or within that observed. PCA or similar multivariate analysis of morphological variation within this population compared to that of all *Fractofusus andersoni* specimens available could also be employed. Even if a comparable number of specimens are not available from a single ‘other’ population (as expressed in line 122-124), the authors can test the hypothesis that including specimens from any other environments/beds/localities should introduce greater variation, providing at least some opportunity to evaluate the possibility of ecophenotypic controls on morphology. This would greatly increase the significance of the findings. Alternatively, if the authors are uncomfortable or cannot do this, they should eliminate all text in the discussion that references ecophenotypic variation (e.g. line 328-330, and all conclusions that follow) as they have not evaluated this factor by their own admission.

This is a very important prompt and we thank the reviewer for raising it. The population of *F. andersoni* is a single population of fossils (and in that sense, we assume broadly similar environmental/depositional regimes over time) but this is not necessarily a single population in a biological sense - time averaging is known and described (Wilby, Kenchington and Wilby 2015; Liu *et al.* 2011) on these fossil surfaces, meaning that individuals of different sizes (more or less developed) may have been exposed to different environmental pressures over the course of their lifetimes. We had not articulated this (at all) in our original submission and have endeavored to make this much clearer.

We do not feel comfortable integrating data collected by others into our dataset because we cannot be sure of their data collection regime - a great number of branches are present at the distal tips, which are often lost first to weathering, and given the only study in which equivalent data (Gehling and Narbonne 2007) is reported is based on specimens from sites with a lot of weathering we are not happy that such a comparison would be meaningful. We have not collected equivalent data from other surfaces ourselves and so don't have any further data to integrate at this time.

We now clarify in the materials and methods "*Fossilised populations of Fractofusus...*" and "*The Fractofusus population examined here may not represent a population in a biological sense because the preserved community has likely undergone time-averaging, as has been documented previously for other Avalonian Ediacaran fossil surfaces*". We have also edited lines 342-344 in the discussion to read "*...core branching pattern is not impacted by environmental variation (insofar as we have examined a time-averaged population of F. andersoni, see⁵¹ also*". We now cite Dunn *et al.* 2018 because they go into this in more detail, looking at the same taxon across different fossil surfaces.

We have now limited reference to ecophenotypic variation in our manuscript; once in relation to a previous growth study on a different taxon which used fossil specimens from multiple sites; once in relation to the results of another, previous study and once to explain why we limit our study to one fossil site. We have removed reference in our conclusion to recovering distinct and diverging lineages in rangeomorphs. We hope these edits and our explanation address the reviewer's concerns.

Line 98-100: Is this an original hypothesis of these authors (that there is little or no difference between the species in the earliest developmental stages)? Or can they reference other work? Do the

results found here offer any new insight? This is seemingly a major contribution of such a detailed study.

This is a novel hypothesis that we suggest, though we note that this uncertainty arises from a lack of early developmental stages of, particularly, *F. misrai*. We did set about trying to collect data to address this question, but decided on reflection to exclude it from this manuscript as we felt the data available for *F. misrai* are insufficient to answer that question. We do have evidence that subsidiary branches (present in *F. misrai* but not *F. andersoni*) are present in specimens as small as 3cm, but we do not know of any smaller specimens to assess to look beyond this. Initial morphometric data show that the shape of the two species is more similar in less developed, smaller specimens, but in the absence of comparably sized specimens of *F. misrai* this comparison is incomplete. We have therefore decided to wait and publish it separately, when more data are available to us.

Lines 109-115 and 183-187: These incomplete specimens are intriguing, and more description is warranted here. First, the authors have not confirmed that these are not the result of folding out of the plane of preservation as stated in lines 185-186. This has been observed in other Ediacara taxa and can be tested by examining whether the incomplete portion in different specimens displays any alignment that can be attributed to current (cf. Evans et al., 2015 P3). While I agree specimens with a midline that extends into these otherwise incomplete sections are unlikely the result of folding – an alternative is that the branches on both sides of this midline are each folded. This may be expected if the midline is a point of attachment and/or more rigid than branches, and the fact that it only occurs in 3 individuals is consistent with inferred low probability. I fully expect alignment to be random, which would then falsify the folded/lifted hypotheses, but this data should be presented.

This is a very good point and we thank the reviewer for raising it. Unfortunately, we do not have all the information required to assess this for all incomplete specimens - our casts or field photos are tightly cropped around the specimen of interest in many cases. However, we do have these data for 9 specimens (including one not previously figured) and observe a range of differences in angles between incomplete *Fractofusus* and nearby strongly-aligned fronds (which are interpreted as having been felled in line with the current), ranging from 0 degrees in one specimen (complete alignment) to 129 degrees. The only complete *Fractofusus* specimen that also shows a filamentous extension for which we have this data does not show alignment to nearby upright fronds. In order to convey this information we have created a new supplementary figure (new Supp. Fig. 2) which

includes orientation and contextual information where available. One additional point to note is that some specimens are incomplete at both distal tips (e.g. Supp. Fig. 1F and in that specimen, another incomplete specimen nearby lies at a different angle to the figured specimen.

We feel that these new data help to falsify the folded/lifted hypothesis.

The authors should be more explicit about their hypothesis for why these specimens are incomplete. They rule out taphonomic explanations, but what caused this? Did development cease in one direction due to some environmental or size factor? Or is this a 'normal' pattern of growth to be expected in all complete specimens? Further elaboration is warranted, even understanding that definitive conclusions may not be possible. Further justification for the stoloniferous interpretation is necessary. Is this the only other non-taphonomic hypothesis that can explain these incomplete specimens?

We can refute the possibility of a developmental or size-related process because the specimens that show these phenomena occur across the observed size range (main text Fig. 2), although we refrain from giving precise measurements in the text because the specimens are so incomplete. This is now explicitly mentioned in our results section *"...terminating abruptly along a line perpendicular to the long axis of the specimen (Fig. 2A–F; Supp. Fig. 1, also noted in F. mirai by ³¹). Such incomplete specimens sample the known size range..."*. Environmental factors such as physical damage, and reproductive strategies including budding/fragmentation, all remain viable possibilities to explain these 'broken' specimens. These are now mentioned in text, copied here (lines 172-181):

"We can rule out the possibility that they represent a deterministic developmental stage because there is no size-range bias visible in our population (Fig. 2). We can also rule out the possibility that these represent partial lifting or folding of specimens because they are not consistently aligned with taxa inferred to have lived upright in the water column, which are very strongly aligned on the Brasier surface, presumably aligned by a unidirectional current³⁷. Other potential explanations include physical damage resulting in lost frondlets, as has been suggested for visibly disturbed branches in other rangeomorph taxa³⁸, or reproductive strategies including budding or fragmentation, if reproductive maturity had been reached by our truncated specimens. We are currently unable to discriminate among these interpretations."

Line 191-195: An alternative explanation for the observed patterns is that the arrangement of first order branches was not tightly controlled during the development of *Fractofusus*, producing specimens with both offset and opposite morphologies. This may be a significant result, suggesting some amount of developmental plasticity in early eumetazoan patterning around a central axis. Discussion of this possibility and any observations to support or refute (e.g., does retro deformation tend to minimize or exaggerate offset?) this hypothesis is warranted.

This is a very useful suggestion and we have now expanded our manuscript to introduce this possibility in the discussion sub-section 'anatomical organisation', where we now state *"Therefore, two possibilities exist: 1) an opposite arrangement was the original branching pattern for the central branches, representing the oldest branches from which the central branching axis diverged laterally, or; 2) the arrangement of first order branches was not tightly controlled during morphogenesis, producing specimens with both offset and opposite morphologies."*

We also include additional text in our discussion to introduce the idea that our data could be compatible with developmental plasticity in early eumetazoan patterning around a central axis. We haven't added text on retrodeformation here as *Fractofusus* specimens are not aligned, and therefore we would not expect any systematic relationship between retrodeformation and *Fractofusus* specimens. We conclude that while we cannot refute this as a competing hypothesis, we find it less likely than the alternative because we only ever observe the opposite branching pair in the middle of the specimen, which would not be expected if growth was plastic around the central axis.

Our discussion now reads (lines 255-262):

*"Our finding of a single pair of bilaterally arranged branches in multiple specimens of *F. andersoni* is compatible with the earliest frondose phase in the *Fractofusus* life cycle being composed of two opposing branches, from which subsequent branches emerged distally and from a central stolon. However, a competing hypothesis is that *Fractofusus* shows developmental plasticity in the arrangement of branches around the central axis. We find this less likely, given the bilateral branching pair are only ever found in the centre of the specimens, which would not be an expectation of this model. Future data from other fossil surfaces will confirm or refute this hypothesis."*

Line 204: Can you provide a reference to figured specimens that exhibit this 'broken' morphology?

We now cite main text Figure 2A-H.

Lines 205-210: What if such damage occurred relatively close to the time of burial?

We have now included the clause “...implying that they were unlikely to have represented degraded necromass at the time of burial (though they could represent specimens damaged shortly prior to burial)”

Line 213-215: This statement could lead to the assumption that no supporting data was published by Taylor et al., 2022 – however, there is supplementary data available (although it does not seem to include the metrics needed to address the hypothesis discussed here). I suggest simply stating that “We find no such evidence in our data to support this interpretation for *F. andersoni*.”

We have made the changes as suggested by the reviewer. We included this clause originally because several key specimens in Taylor *et al.* 2022 - which their hypothesis relies on - are not present in the supporting data published by that team. When this was raised with them, in December 2022 Taylor and colleagues told one of us that they were working on an erratum to include those missing data. Two years later, no erratum has been published.

Line 237-242: Why is the hypothesis of 2 opposing first order branches supported over any other number – certainly the y-intercept of figure 3A would be much greater than 2 – suggesting many more initial branches? Additionally, the lack of consistent labelling in figure 3 leads to some confusion. Please add more explicit indication that the y-axis in figure 3A does not begin at 0. Why is zero labelled in the y-axis of some panels and not others, or for the x-axis in any panel? Convention suggests that the origin of each graph begin with (0,0) and be labelled accordingly.

We thank the reviewer for spotting this oversight in our plots - all that did not intercept at 0,0 have been replotted to do so (Fig. 3).

On re-reading our text, we agree that there was not enough nuance or justification. The only other rangeomorph for which these detailed growth data had been collected - *Charnia* - shows a period early in the developmental history of the organism with rapid differentiation of branches, which we

do not capture in *Fractofusus* with the specimens available to us. Early rapid differentiation may account for the unexpectedly large number of first order branches in the smallest *Fractofusus* specimens measured. It is also possible that a number of branching pairs differentiated simultaneously, however such a hypothesis provides no explanation for our observations of a single pair of opposite branches only at the centre of the frond. Given this, we prefer our original hypothesis, but have added text noting that this alternative hypothesis exists and we cannot rule it out (lines 231-239): “*It is possible that Fractofusus went through a differentiation-dominant phase of growth very early in its developmental sequence⁹, with the smallest reported Fractofusus in this study being 7.8 mm in length and possessing 18 first order branches in total. Our interpretation of opposite branches as a likely original biological feature may suggest that Fractofusus began the frondose stage of its life cycle with just two, opposing, first order branches. An alternative scenario could see a series of branching pairs differentiating simultaneously, but that provides no explanation for the presence of two, opposing first order branches at the centre of the organism as opposed to any other position. Therefore, we prefer the former hypothesis based on our data but note that we cannot rule out the latter.*”

Morphogenesis: Can the authors provide a figure that documents changes in *F. andersoni* through the developmental stages observed. Something like Figure 4 of Dunn et al., 2021 Science advances?

We have now removed the original Figure 5 and replaced it with a new figure showing the developmental stages of *F. andersoni*.

Line 261-289: The arguments of this section are difficult to follow. This study details a single population of a single species, the conclusions drawn here between other rangeomorphs are therefore incomplete and poorly defined. Much of the argument seems to simply state that different fronds have different morphology. Figure 5 does not add to this, displaying 3 different fronds but failing to “rationalize variation between them” (as stated in the abstract) given the arrows suggest changes could proceed in any direction? This figure may be redundant since various rangeomorph morphologies are displayed in Figure 1. The model presented seems to hinge around changes to the main axis, but in one of three key taxa discussed (*Bradgatia*) this is cryptic and so must be inferred. If morphology of the main body changes it follows that the main axis changes too, what evidence is there that this is the controlling factor? This section is likely better suited to a separate manuscript dedicated to fully elaborating these changes and including observations from a significant proportion

of Rangeomorph taxa, rather than the few examples selected and the incomplete discussion that follows.

We compare *Fractofusus* to all rangeomorph taxa for which there is information about developmental pattern, so in that sense we view our discussion of rangeomorphs as complete, but we understand the reviewer's point and have tried to make our aims clearer in this section. Once we have established similarities and differences in how these different fronds grew, we propose a model that accounts for morphological variation between these taxa. We agree with the reviewer that we do not polarize this change, which would require a phylogenetic analysis that is beyond the scope of the current study.

We also felt that, on re-reading, it was not clear how our study impacted upon hypotheses of homology between taxa that underpin our model to account for morphological transitions between rangeomorphs; we have rewritten this section in order to highlight this, copied below (lines 280-298):

"Dunn et al. 2021⁹ proposed a scheme to find homology between disparate rangeomorph taxa and suggested that the entire frond of Charnia was homologous to a single first order branch of Fractofusus. However, that study did not go further in attempting to find homology between the non-frondose components of rangeomorph bodyplans and so it was not possible to rationalise the disparate rangeomorph bodyplans. Our novel data on Fractofusus help resolve this: recent work has shown that in at least some uniterminal fronds, potentially including Charnia, individuals of the same species and of significant size (indicating a role extending beyond simple reproduction) had filamentous connections ^{34,36}, similar in expression to the central stolon of Fractofusus that we present here. This observation provides an obvious way to rationalise between the bodyplans of these rangeomorphs, with differences in gross morphology explained by simple compression and/or extension of the fundamental axis (that from which the lowest [thus largest in size] order of branches derive), with individual fronds differentiating distally along a stoloniferous extension. This model is easily applicable to most other rangeomorph taxa, except bush-like forms including Bradgatia. There, the fundamental axis is cryptic but it must represent the axis along which branches differentiate from the holdfast. Following Dunn et al. 2021⁹ each individual folium in Bradgatia is homologous with the entire Charnia frond, or a single first order branch in Fractofusus (as it constitutes the largest

branching unit that is independent of all others). A mechanism by which to generate, or lose a bush-like morphology is therefore suggested to be contraction of a lateral branching axis.”

Line 281: Lowest order (thus largest in size)?

Edited in line with the reviewer’s suggestion.

Line 293-295: Provide references to this claim. Is this true of modular taxa, or just colonial forms? Would you consider *Fractofusus* colonial?

We have now added references to this. Whether *Fractofusus* is colonial or not is a good question! If we extend the homology scheme we propose, then *Fractofusus* would represent many individual frondlets arranged along a stolon, therefore a colony. In other rangeomorphs, filamentous strands connect fronds of substantial size (i.e. presumed to be ‘adult’), indicating perhaps that they are not necessarily exclusively used in reproduction (although we see no evidence of this in *Fractofusus* specifically) and raising the possibility that these, too, represent colonies. In expanding the section on potential homologies in response to the comment above, we now include information on whether *Fractofusus* and other rangeomorphs are or are not colonial, copied below (lines 300-308):

*“This model equates single first order branches in *Fractofusus* with multiple, interconnected uniterminal fronds that have been interpreted as clonal or colonial³⁴ and therefore could suggest that *Fractofusus* itself may be a clonal colony. If this is the case, *Fractofusus* would show extremely high levels of colony integration, with a bodyplan that remained static across its multi-million year stratigraphic range⁵. This model is not incompatible with our morphogenetic data which shows that the position of the branches along the distal-proximal axis scales linearly with length in the examined size range, unlike in *Charnia* where there is a second order polynomial relationship between branch length and position in the apical-basal axis⁹ indicating tighter spatial regulation of branching growth than in *Fractofusus*.”*

Line 355: Double negative, consider revising.

The hypothesis is of minimal developmental regulation and we do not find support for it. We have chosen to leave the text as-is.

Line 366-365: This statement (and similar statements elsewhere – e.g., lines 28-29 of the abstract) is not supported by the main body of the text. Animals today exhibit variable “developmental regulatory capacity” (e.g. number of Hox genes), where does *Fractofusus* fit within this range? More importantly, this statement oversimplifies the distinction between possessing the gene regulatory tools necessary to produce complex morphologies (which likely extends well beyond metazoan lineages) and the co-option of those tools that results in their expression. The current manuscript documents that growth in *Fractofusus* was modular and highly regulated – an important find and one worth publishing. However, this is certainly less complex than the regulatory pathways that need to be established to produce differentiated body regions, organs, appendages, etc. found later in the fossil record.

We have altered the final sentence of the abstract, in line with comments from reviewer two which we hope also addresses this comment:

“Our results imply that complex developmental regulatory machinery was already being utilised during the late Ediacaran in the earliest-diverging eumetazoan taxa represented in the fossil record.” *Fractofusus* is unusual amongst rangeomorphs in that it is entirely constructed of a central filamentous thread and frondlet branches; most other taxa possess features that you would certainly describe as differentiated body regions and could conceivably call organs, including a holdfast disc (e.g. *Charnia*, *Primocandelabrum*, *Bradgatia*), or a central upright stalk/stem (*Avalofractus*, *Hylaecullulus*, *Plumeropriscum*) which are distinct in their construction from the frondlets they host. We have expanded our discussion of developmental complexity in line with this information. We chose to study the growth of *Fractofusus* because it is so distinct from other rangeomorphs, but the reviewer makes a very reasonable point, which we hope we have now addressed in lines 378-391.

“Living eumetazoans are characterised by highly complex spatially and temporally controlled organ systems, including guts, muscles and nervous systems. Rangeomorphs appear to have diverged before the appearance of some or all of these key synapomorphies, but nonetheless display differentiated tissues and perhaps organs. Fractofusus is unlike most other rangeomorphs in that it is constructed entirely of frond branches and a stoloniferous central axis, whereas other taxa possess structures including holdfast discs, stems and stalks which differ in their construction to the frond itself. The functional morphology of rangeomorphs remains controversial^{62,63} but the morphological distinction between the frond, holdfast, stalk and stem is compatible with their interpretation as

organs, in the sense that these represent differentiated anatomical units specialised to perform specific functions. Therefore, the fundamental regulatory pathways that establish body regionalisation in space must have already been established and co-opted into these roles deep within the eumetazoan stem-lineage, pathways that were elaborated in the construction of crown-group eumetazoan organ systems.”

Reviewer #2 (Remarks to the Author):

In this manuscript, the authors examine growth and morphogenesis in the Ediacaran taxon *Fractofusus* on a single bedding plane. *Fractofusus* is one of the more iconic and generally well-known taxa of the suite of Ediacaran fossils representing the oldest of the Ediacaran fossils – commonly referred to as the Avalon assemblage. *Fractofusus* is part of a group of taxa that exhibit self-similar branching that are broadly referred to as Rangeomorphs sharing a similar branching mode and these morphotypes characterize these oldest Ediacaran assemblages.

The authors demonstrate effectively that *Fractofusus* had highly regulated growth. This is a new and important contribution. Detailed studies of growth of the enigmatic Ediacara biota are few but notably include the similarly enigmatic *Dickinsonia*. While there have been a number of studies of growth in the Rangeomorphs – more than any other Ediacaran “clade”, this study represents a new approach and insight. A single bedding plane eliminates a number of potential environmental and taphonomic issues. Furthermore, the authors address morphogenesis in a novel but detailed manner.

The results are clear cut and I think very straight forward. While this does not radically change our interpretation of *Fractofusus* or Rangeomorphs, it clearly shows that we need to regard them from a very different perspective than just that they are “weird and different” and that there is no way to constrain them on the tree of life. That’s important not just for these taxa but for understanding the early evolution and evolutionary experimentation among early forms. The documentation of highly regulated growth amongst Earth’s earliest multicellular heterotrophs is important.

We thank the referee for their kind and supportive comments!

The authors push the importance of teratologies (line 348 onwards) which actually detracts from the strength of the paper. At best, these are rare. Many would argue with the Dickinsonia interpretation and the authors do not cite a rangeomorph example. This feels like a strawman and is not necessary. They have adequately demonstrated highly regulated growth – teratology is not key and pulls the argument down.

While people may argue about the ultimate cause of the abnormalities in *Dickinsonia* or indeed rangeomorphs (we did cite an example here in our original submission, that of eccentric branching described in *Hylaecullulus* and other multifoliate rangeomorphs), it is not controversial that the impact is a change in the pattern of morphogenesis - there is no other way that aberrant morphology could be generated. Therefore, we have changed the word 'teratology' throughout so as not to imply congenital abnormality as the ultimate cause.

Developmental abnormalities are rare in Ediacaran macrofossils, but they remain informative regarding the regulatory capacity of these organisms and therefore provide an independent test of our model of morphogenesis. Developmental abnormalities are used in the study of living organisms in the same way, and have resulted in seminal discoveries, e.g. cell cycle regulation in budding yeast or body axis regulation in the fruit fly. We have therefore chosen to keep this discussion, but have included an additional justifying sentence (lines 369-371): *"The identification of abnormalities therefore provide an independent test of our model of morphogenesis which is based on specimens that have developed normally."*

I also don't think that the authors need to emphasize the Precambrian-Cambrian boundary. I know that in some circles, this is a "put your finger on it" evolutionary moment. But nowadays most don't see it that way. I think a better way to approach would be to put this all in the context of early animal evolution not tying it to a stratigraphic boundary that is really just about the appearance of one trace fossil. The main part of the Cambrian radiation was much later – but a great deal of evolution happens between the Avalon assemblage and the Cambrian radiation. Putting this in a much broader context would be useful to readers from a broader audience.

This is a really useful suggestion and we have implemented it - thank you!

In the abstract we now conclude with: *“Our results imply that complex developmental regulatory machinery was already being utilised during the late Ediacaran in the earliest-diverging eumetazoan taxa represented in the fossil record.”*

In the conclusion we now write: *“Our data imply a morphogenetic strategy that was highly regulated, and demonstrate that the most ancient eumetazoan fossils known already manifest evidence of complex developmental regulation.”*

We have also removed reference to the Ediacaran-Cambrian transition from the introduction.

Line 42 – I wouldn’t say striking given the diversity of morphology in animals – in the end, the variation is impressive given that they are much of a muchness – that is all grow in a self-similar way.

We have changed this to read *“they are morphologically varied”*.

Line 54 – typo – Dickinsonia not Dicksinsonia

We thank the reviewer for spotting this typo.

125 – assume that this means that they were just smaller?

Yes, we have clarified this in text now as well.

329 – Highly regulated growth does not mean that they were not malleable. It is hard to really say without comparisons to populations on other beds. There are certainly organisms with highly regulated growth that are indeed malleable.

The reviewer is correct, thank you for picking this up. We have now rewritten this sentence to make it clear that we are referencing previous work:

“Rangeomorph bodyplans were not defined by environmental parameters (contra ^{24,57}) and do not appear particularly malleable^{9,51}.”